# Is Femoral Neck System a Valid Alternative for the Treatment of Displaced Femoral Neck Fractures in Adolescents? A Comparative Study of Femoral Neck System versus Cannulated Compression Screw

**DOI:** 10.3390/medicina58080999

**Published:** 2022-07-27

**Authors:** Yunan Lu, Federico Canavese, Guoxin Nan, Ran Lin, Yuling Huang, Nuoqi Pan, Shunyou Chen

**Affiliations:** 1Department of Paediatric Orthopaedics, The Third Clinical Medicine College of Fujian Medical University, Fuzhou Second Hospital Affiliated to Xiamen University, 47th Shangteng Road of Cangshan District, Fuzhou 350007, China; lyn5718@163.com (Y.L.); linran312@163.com (R.L.); yulinghuang01@163.com (Y.H.); pnq8798@163.com (N.P.); 2Lille University Centre, Department of Paediatric Orthopaedic Surgery, Jeanne de Flandre Hospital, Rue Eugène Avinée, 59000 Lille, France; canavese_federico@yahoo.fr; 3Department of Orthopaedics, Children’s Hospital of Chongqing Medical University, Yuzhong District Zhongshan 2 Road 136#, Chongqing 400014, China; ngx1215@163.com; 4Fujian Provincial Clinical Medical Research Center for First Aid and Rehabilitation in Orthopaedic Trauma, Fuzhou 350007, China

**Keywords:** femoral neck system, cannulated screws, femur neck fractures, internal fixation

## Abstract

*Background and Objectives*: The femoral neck system (FNS) is a new minimally invasive internal fixation system for femoral neck fractures (FNFs), but its use has not been reported in adolescents. The aim of this study was to compare the clinical and radiographic outcomes of displaced FNF in adolescents treated with FNS or a cannulated compression screw (CCS). *Materials and Methods*: A retrospective study of 58 consecutive patients with displaced FNF treated surgically was performed; overall, 28 patients underwent FNS and 30 CCS fixation. Sex, age at injury, type of fracture, associated lesions, duration of surgery, radiation exposure, and blood loss were collected from the hospital database. The clinical and radiographic results, as well as complications, were recorded and compared. *Results*: The patients were followed up for 16.4 ± 3.1 months on average after index surgery (range, 12 to 24). Consolidation time among patients treated with FNS was significantly lower than those managed by CCS (*p* = 0.000). The functional scores of patients treated with FNS were significantly higher than those managed by CCS (*p* = 0.030). Unplanned hardware removal in patients treated with FNS was significantly lower than in those managed by CCS (*p* = 0.024). *Conclusions*: FNS has a lower complication rate and better functional outcome than CCS. It may be a good alternative to treat femoral neck fractures in adolescents.

## 1. Introduction

Femoral neck fractures (FNFs) in pediatric age are rare, accounting for less than 0.5 percent of all fractures in children and adolescents; the peak incidence is found in male children aged 11 to 14 years. Despite their rarity, however, they have a very high complication rate, such as avascular necrosis of the femoral head (AVN), femoral neck shortening, coxa vara, premature closure of the proximal femoral growth cartilage, and nonunion [1,2,3].

Internal fixation is still the best option for displaced FNF in adolescents. However, the best fixation system remains in dispute. According to a recent study by Wang et al., the cannulated compression screw (CCS) is the most frequently used system of fixation in pediatric FNF in the past decade, applied in up to 80.8% of cases [4].

Stoffel et al. introduced a new minimally invasive internal fixation system for FNF called the femoral neck system (FNS) [5] (Figure 1). Biomechanical studies showed that the stability provided by FNS was similar to that of a dynamic hip screw (DHS) and hip dynamic blade screws, and it was better than that of CCS [5,6]. To date, several studies have reported on the efficacy of FNS in adults with FNF [7,8,9]. However, no one has investigated the use of FNS in adolescents.

The aim of this retrospective study was to evaluate the clinical and radiographic outcomes of displaced FNF in adolescents managed by FNS or CCS and to compare the two treatment options.

## 2. Materials and Methods

### 2.1. Patients

After obtaining Institutional Review Board (IRB) approval (No. *2022007*), a retrospective comparative study was performed on 58 consecutive pediatric patients with FNF treated with FNS or CCS from March 2017 to February 2021. The demographic characteristics and clinical profiles, including age at the time of injury, sex, side, time from injury to surgery, mechanism of injury, type of fracture according to the Delbet–Colonna and Pauwels classifications, and initial displacement, were recorded.

The inclusion criteria were as follows: (1) a confirmed diagnosis of isolated, closed FNF; (2) age between 10 and 20 years at the time of injury; (3) surgical treatment with FNS or CCS within 14 days of the injury; (4) follow-up >12 months; and (5) complete clinical and radiological data.

The exclusion criteria were as follows: (1) concomitant neuromuscular, metabolic, or genetic conditions, and pathological fractures; (2) slipped capital femoral epiphysis or FNF combined with greater or lesser trochanter fracture; (3) PFMI grade 6 or above [10]; (4) incomplete medical records and/or imaging data, and a follow-up of less than 12 months.

### 2.2. Surgical Procedure

All surgical procedures were performed under general or spinal epidural. All surgical procedures were performed by the same experienced pediatric orthopedic surgeon (S.C.). The fracture was reduced with the patient supine on an orthopedic traction table. If closed reduction failed after 2 to 3 attempts, open reduction was performed.

### 2.3. Femoral Neck System

Before making a 5 cm longitudinal incision under the greater trochanter, the fractured limb was slightly abducted and internally rotated. The soft tissues were subsequently stripped from the bony plane (lateral aspect of the proximal femur) in order to facilitate hardware placement. After temporarily stabilizing the fracture by inserting a Kirschner wire, a second Kirschner wire was placed as a central guide wire using a 130° angled guide, whose correct position was confirmed using fluoroscopy. The optimal implant length was measured with a scaling device. We tried to choose an appropriate length to spare the growth of the proximal femur epiphysis; violation of the growth plate was necessary only in high and unstable fractures. At this point, the bolt preassembled with the plate was slid over the central guide wire into the preformed hole. Next, the anti-rotation screw was inserted after extracting the guide wire. Finally, the locking screw was placed in the distal hole of the plate, and the stability of the system was tested by moving the hip in all directions. The position of the SNF (DePuy Synthes, Switzerland) and the quality of fracture reduction (Figure 1 and Figure 2) were checked using fluoroscopy.

### 2.4. Cannulated Compression Screw

Two or three incisions of about 0.5 cm each were made to allow the insertion of 2 or 3 guide pins, depending on the age of the patient and the width of the femoral neck, into the femoral head along the longitudinal axis of the femoral neck; insertion was performed under fluoroscopic control. After the guide wires were placed in the appropriate position, cannulated screws were inserted. The proximal femur epiphyseal growth plate was not violated unless the fracture was unstable and had a short proximal fragment. The fracture fragments were considered fully compressed if all threads were above the fracture line (within the proximal fragment) on the AP and lateral postoperative radiographs.

### 2.5. Perioperative Management

All patients received prophylactic antibiotic therapy 30 min before the start of surgery. Once surgery was completed, patients were encouraged to begin isometric quadriceps femoris contraction exercises and to perform active flexion and extension of the ankle and knee joints. Partial weight-bearing training was implemented according to the recovery of the affected limb. Full weight bearing was allowed once bone consolidation was achieved, usually 3 months after surgery. The first radiographic examination was performed within 3 days after surgery. The following radiographic examinations were performed once a month during the first 6 postoperative months and every 3 months thereafter. Functional assessment of the hip was performed at the last follow-up visit, no less than 12 months after surgery. A computed tomography (CT) or magnetic resonance imaging (MRI) scan to look for fracture nonunion or necrosis of the femoral head was performed in all patients with persistent hip pain. All measurements were performed using the Picture Archiving and Communication Systems (PACS; GE Healthcare, Chicago, IL, USA). Two experienced pediatric orthopedic surgeons (Y.L. and R.L.) measured all parameters independently, and the mean values were used for the statistical analysis.

### 2.6. Radiographic Evaluation

Two experienced pediatric orthopedic surgeons (Y.L. and R.L.) measured all parameters independently, and the mean values were used for the statistical analysis.

To identify adolescent patients, the PFMI was used. Adolescents with PFMI ≥ 6 were not included in the analysis [10].

The Delbet–Colonna system was used to classify all fractures: type I, transepiphyseal fracture; type II, transcervical fracture; type III, cervicotrochanteric fracture; and type IV, intertrochanteric fracture [11].

Pauwels classification calculates the angle between the fracture line of the distal fragment and the horizontal line, as follows: type I: <30°; type II: 30°–50°; type III: >50° [12].

Song et al. [13] and Wang et al. [14] classification system was used to evaluate initial fracture displacement: type I, incomplete fractures without translation or mild angulation <30°; type II, complete fractures with any amount of translation or angulation <50°; and type III, complete fractures with any translation or angulation >50°.

Song et al. [13] classification was used to evaluate the quality of reduction: anatomical, with no displacement or angular deformity; acceptable, displacement < 2 mm or angular deformity within 20° of the normal neck shaft angle on anteroposterior and axial radiographs; unacceptable, displacement of >2 mm or angular deformity of >20° on anteroposterior or axial radiographs.

Functional evaluation was performed using Ratliff’s criteria [15] considering pain, ROM (good, fair, and poor), activity, and femoral neck morphology on radiographs, which was excellent, good, or poor.

Ratliff’s classification [15] was also used to assess the presence and severity of femoral head necrosis: type I (necrosis of the whole femoral head), type II (partial necrosis of the femoral head), and type III (an area of necrosis from the fracture line to the physis).

Neck shortening was evaluated according to Zlowodzki’s method [16]. The neck of the femur was considered to be shortened in the presence of a shortening, on radiographs, of the distance between the tip of the femoral head and the caudal end of the lesser trochanter greater than 5 mm on the affected side compared with the contralateral, healthy side. Premature physeal closure was defined as 50% or more linear closure of the physis [17].

Coxa vara was defined as neck-shaft angles < 120°.

Nonunion was defined as implant breakage, loss of reduction, or persistence of a visible fracture line ≥ 6 months after the index procedure [18].

The material was removed prematurely (nail retreat) when the screw head slipped out more than 5 mm from the cortex of the lateral side of the femoral shaft, causing hip discomfort.

### 2.7. Statistical Analysis

Data were analyzed using the IBM SPSS statistical package version 22.0 (IBM Corporation, Armonk, NY, USA). The Shapiro–Wilk test was first used to determine whether the data fit a normal distribution. Age, time from injury to surgical treatment, perioperative blood loss, and consolidation time were normally distributed and expressed as the mean, range, and standard deviation, and two independent sample T tests were used for comparisons between the two types of treatments. Categorical parameters are expressed as frequencies and percentages. Statistical analysis was performed using the chi-square test or Fisher’s exact test for categorical variables. Statistical significance was defined by *p* values of *p* < 0.05.

## 3. Results

A total of 58 patients met the inclusion criteria. The patients were divided into the FNS group (*n* = 28) and CCS group (*n* = 30) based on the type of fixation (Figure 2 and Figure 3). Table 1 summarizes the demographics of the patients (Table 1); no significant differences in demographic characteristics, such as age, sex, mechanism of injury, type of fracture, or initial displacement, were found between the two groups (*p* > 0.05).

Table 2 summarizes the operative characteristics of the FNS and CCS groups (Table 2). The intraoperative blood loss of patients treated with FNS (44.3 ± 7.8 mL; range, 30 to 70) was significantly higher than that of patients treated with CCS (16.7 ± 4.7 mL; range, 12 to 25) (*p* = 0.001). There was no significant difference in the reduction method, time from injury to surgical procedure, fluoroscopy time, or quality of fracture reduction between the two groups of patients (*p* = 0.928).

Patients treated with FNS and CCS were followed up for an average of 16.3 ± 2.0 months (range, 12 to 24) and 17 ± 2.5 months (range, 12 to 24), respectively (*p* = 0.232). The FNS group had a shorter fracture consolidation time (10.5 ± 1.5 weeks; range, 8 to 14) than the CCS group (14 ± 2.2 weeks; range, 10 to 20) (*p* = 0.000). Similarly, patients treated with FNS had higher functional scores (excellent: good: poor = 15: 10: 3) than patients managed using CCS (excellent: good: poor = 7: 13: 10) (*p* = 0.030). Complications such as AVN (4/14.3% vs. 8/26.7%) (Figure 4) and neck shortening (2/7.1% vs. 6/20%) were lower in FNS than in CCS patients, but without statistical difference (*p* = 0.245 and 0.256, respectively); unplanned hardware removal (0/0% vs. 5/16.7%) (Figure 3) was significantly lower in FNS than in CCS patients (*p* = 0.024) (Table 3).

## 4. Discussion

FNF in adolescents is a challenging injury to treat due to the complex vascular anatomy of the femoral neck and the high-energy traumatic mechanism responsible for the fracture and its displacement. Anatomical reduction of the fractures and stable internal fixation, taking into account the vascularization of the proximal femur, are essential elements of optimal treatment.

In recent decades, CCS has been the preferred fixation for FNF in adolescents due to its minimally invasive technique, reduced cost, and its ability to adequately address most femoral neck fractures [19]. Although the locking compression pediatric hip plate (LCP-PHP) was originally designed for intertrochanteric and subtrochanteric osteotomies, they have also been used for the treatment of pediatric FNF [20]. Joeris et al. [20] and Chen SY et al. [21] found that LCP-PHP can provide better angular stability, better resistance to rotation and shear forces, and better functional outcomes than CCS. However, the use of such plates often requires a large incision, resulting in greater surgical damage and perioperative blood loss.

The recently introduced FNS has the minimally invasive properties of CCS and the stability of LCH-PHP fixation. In addition, it can provide compression, maintains the femur neck length, and promotes bone healing [5]. Our study showed that the consolidation time in patients treated with FNS was significantly shorter than in patients managed by CCS (*p* < 0.05), while functional outcome as per Ratliff’s criteria was significantly better (*p* < 0.05). This finding was similar to the results reported by He et al., Hu et al., and Zhou et al., conducted in adult patients with FNF [7,8,9].

The rate of nail retreat was significantly lower in patients treated with FNS than in patients managed with CCS (*n* = 5, 16.7%) (*p* < 0.05). The FNS has a 20 mm sliding compression space to facilitate anatomical reduction and compression, resulting in rapid bone healing. The 20mm sliding space is a unique part of the power rod sleeve, which controls the direction at 130 degrees of sliding pressure, thus increasing certain internal resistance and reducing shortening of the femoral neck. In contrast, CCS does not have this advantage. The stability of fractures depends upon the fracture morphology. For example, Pauwels type III FNF is more likely to slip, and the femur neck is more likely to shorten with implant retreat and skin irritation. CCSs often require unplanned hardware removal as a result of skin irritation (Figure 3). However, FNS is more stable and is less likely to cause discomfort (Figure 2).

Interestingly, we found a lower rate of AVN (4/14.3%) in patients treated with FNS than in patients managed using CCS (8/26.7%), although the difference was not statistically significant. Previous reports found that age, fracture type, amount of dislocation, and quality of reduction were predictive factors for AVN; however, fixation methods were rarely investigated [4,13,22].

However, too large an internal fixation can disrupt hemoperfusion of the femoral head, as reported by Wang et al. [14]. The FNS consists of a bolt and an anti-rotation screw with an angle of 5°. The size of the proximal part of the FNS is smaller than that of three CCSs (Figure 4) and reduces the risk of damaging the vasculature of the femoral head epiphysis. In addition to the advantages of smaller size, the FNS is also easier to insert. Once the central guide wire is placed, the rest of the implant can be completed using the guide instrument, resulting in reduced risk of iatrogenic injury. In contrast, optimal placement of CCSs may need multiple attempts and position adjustments, which risk damaging blood supply and causing AVN. However, the possibility that new internal fixation may reduce the frequency of AVN needs to be constantly explored and tested.

Regarding premature epiphyseal closure, we have not yet found any obvious difference (*p* > 0.05). We try to preserve the epiphyseal plate in both forms of internal fixation unless the fracture is unstable. In our series, fluoroscopy time and length of surgery were comparable between the two groups of patients.

There were some limitations in the analysis of our results. These limitations could be mainly attributed to the retrospective nature of the study, the small number of patients, and the relatively short follow-up time (<2 years). Moreover, the treatment strategy (FNS or CCS) was selected according to the surgeon’s preference and was not randomized. Despite the limitations, this is the first study to evaluate the outcome of FNF in adolescents treated with FNS versus CCS. A larger-scale, prospective randomized controlled study with long-term follow-up is necessary to comprehensively evaluate the effectiveness and safety of the FNS method.

## 5. Conclusions

FNS appears to be a safe and effective fixation system for the treatment of displaced FNF in adolescents. FNS has a lower complication rate and a better functional outcome than CCS.

## Figures and Tables

**Figure 1 medicina-58-00999-f001:**
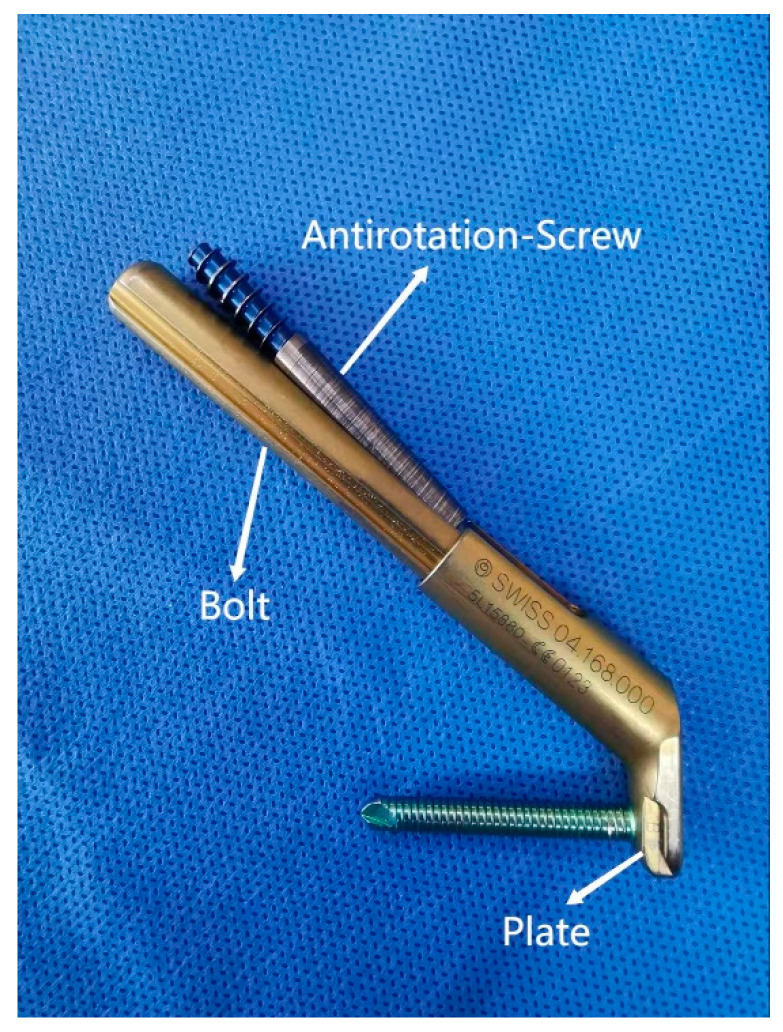
The FNS consists of three parts: the plate and locking screw in the angular stable structure (neck shaft angle = 130°); the bolt (diameter = 10 mm); the anti-rotation screw (diameter = 6.4 mm, branched out from the base of the bolt at an angle of 5°).

**Figure 2 medicina-58-00999-f002:**
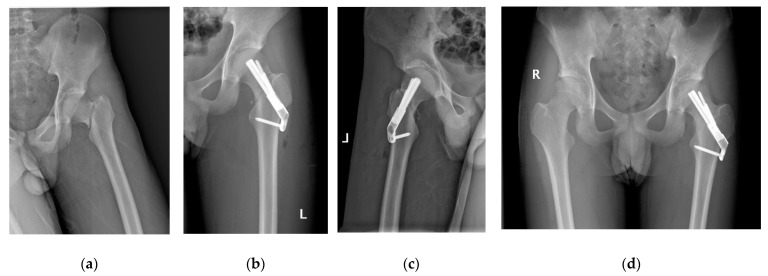
14-year-old boy with left FNF treated with FNS. (**a**) Preoperative radiographs showing Delbet-III, Pauwels-III fracture; (**b,c**) postoperative radiographs; (**d**) outcome at 18 months postoperation.

**Figure 3 medicina-58-00999-f003:**
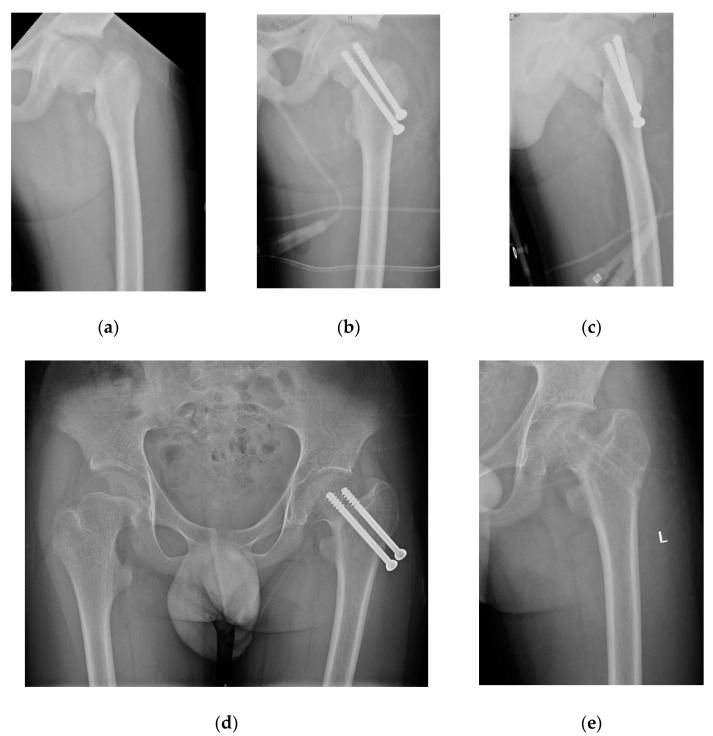
A 13-year-old boy with left FNF treated by 2 CCS. (**a**) Preoperative radiographs showed Delbet-II, Pauwels-III fracture; (**b,c**) postoperative radiographs; (**d**) outcome at 18 months postoperation; (**e**) unplanned hardware removal.

**Figure 4 medicina-58-00999-f004:**
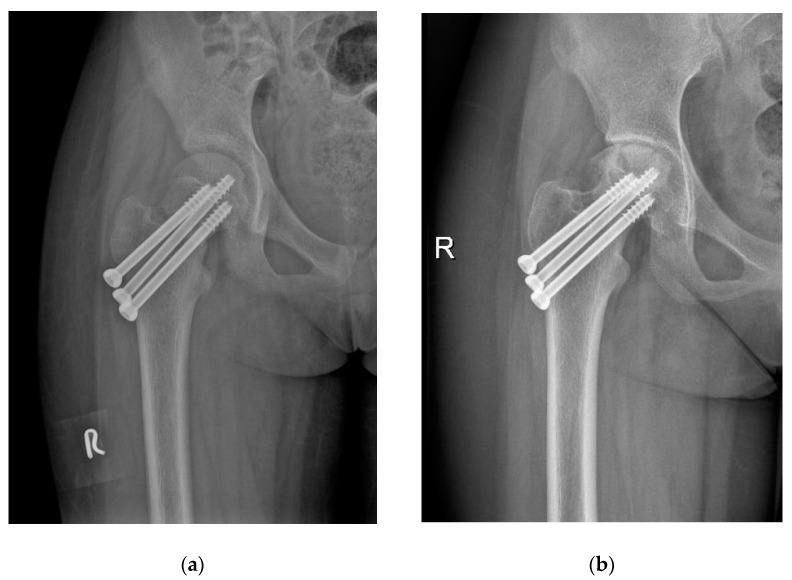
12-year-old girl with left FNF treated by 3 CCS. (**a**) Postoperative radiograph; (**b**) outcome at 12 months postoperation (AVN and narrowing joint space).

**Table 1 medicina-58-00999-t001:** Demographic characteristics of the patients.

Variables	FNS (*n* = 28)	CCS (*n* = 30)	Statistic (t/χ^2^)	*p* Value
Age (years)	14.5 ± 1.6	14.3 ± 1.5	0.394	0.695
Sex				
Male	19 (67.9%)	22 (73.3%)	0.210	0.647
Female	9 (32.1%)	8 (26.7%)		
Side				
Left	16 (57.1%)	13 (43.3%)	1.105	0.293
Right	12 (42.9%)	17 (56.7%)		
Time from injury to surgery (days)	3.2 ± 1.5	3.6 ± 1.9	−0.869	0.389
Mechanism of injury				
Traffic accident	14 (50%)	10 (33.3%)	4.088	0.129
Fall from height	6 (21.4%)	14 (46.7%)		
Sports injury	8 (28.6%)	6 (20%)		
Delbet–Colonna Type				
Delbet-II	20 (71.4%)	18 (60%)	0.837	0.360
Delbet-III	8 (28.6%)	12 (40%)		
Pauwels type				
Pauwels-I	2 (7.1%)	8 (26.7%)	4.402	0.111
Pauwels-II	18 (64.3%)	13 (43.3%)		
Pauwels-III	8 (28.6%)	9 (30%)		
Initial displacement				
Type I	2 (7.1%)	4 (13.3%)	0.914	0.633
Type II	14 (50%)	16 (53.3%)		
Type III	12 (42.9%)	10 (33.3%)		

FNS: femoral neck system; CCS: cannulated compression screw.

**Table 2 medicina-58-00999-t002:** Comparison of operative characteristics.

Variables	FNS (*n* = 28)	CCS (*n* = 30)	Statistics (t/χ^2^)	*p* Value
Reduction method				
CRIF	20 (71.4%)	24 (80%)	0.581	0.446
ORIF	8 (28.6%)	6 (20%)		
Operation time (min)	53.9 ± 10.6	50.0 ± 7.6	1.616	0.112
Blood loss (mL)	44.3 ± 7.8	16.7 ± 3.1	17.932	0.001
Fluoroscopies (*n*)	16.9 ± 4.7	16.0 ± 4.2	−0.090	0.928
Reduction quality				
Anatomical	19 (67.9%)	21 (20.7%)	0.432	0.806
Acceptable	7 (25%)	8 (26.7%)		
Unacceptable	2 (7.1%)	1 (3.3%)		

FNS: femoral neck system; CCS: cannulated compression screw.

**Table 3 medicina-58-00999-t003:** Comparison of clinical effects and complications.

Variables	FNS (*n* = 28)	CCS (*n* = 30)	Statistics (t/χ^2^)	*p* Value
Follow-up time (months)	16.3 ± 2.0	17.0 ± 2.5	−1.208	0.232
Consolidation time (weeks)	10.5 ± 1.5	14.0 ± 2.2	−7.054	0.000
Functional evaluations				
Excellent	15 (53.6%)	7 (23.3%)	7.009	0.030
Good	10 (35.7%)	13 (43.3%)		
Poor	3 (10.7%)	10 (33.3%)		
Complications				
Femoral head necrosis	4 (14.3%)	8 (26.7%)	1.353	0.245
Neck shortening	2 (7.1%)	6 (20%)	2.013	0.156
Nail retreat	0 (0%)	5 (16.7%)	5.107	0.024
Premature epiphyseal closure	3 (7.1%)	3 (6.7%)	0.008	0.929
Coxa vara	1 (3.6%)	3 (10%)	0.932	0.334
Nonunion	0 (0%)	1 (3.3%)	0.950	0.330

FNS: femoral neck system; CCS: cannulated compression screw.

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
