# Peer review of "Is Femoral Neck System a Valid Alternative for the Treatment of Displaced Femoral Neck Fractures in Adolescents? A Comparative Study of Femoral Neck System versus Cannulated Compression Screw"

_medicina, 2022, doi:10.3390/medicina58080999_

Round 1

Reviewer 1 Report

The present paper is interesting and you have done a good job, obtaining and using for this study a big number of participants, knowing that this pathology is not very common.

It is a well written paper and it needs some small corrections. 

Check and correct, please:

"Song et al.Error! Bookmark not defined. classification was used to evaluate quality...."

In discussions are some different styles/number.

Author Response

The present paper is interesting and you have done a good job, obtaining and using for this study a big number of participants, knowing that this pathology is not very common.

It is a well written paper and it needs some small corrections. 

Check and correct, please:

"Song et al.Error! Bookmark not defined. classification was used to evaluate quality...."

-Thanks for pointing this out. The reference has been corrected as suggested.

In discussions are some different styles/number.

-Thank you. The font style and size of the discussion section have been revised. They are now uniform.

Reviewer 2 Report

- No abbreviations in the abstract

- insert exact p values not only <0.005

- 2.6. Error?

- Who performed the radiological assessment? Inter -intraobserver?

- Any remaining complications or remaining negative effects (ROM, etc.)?

poor or only good results and remaining difficulties should be described more precisely after this relatively short FUP since recommended Follow-up is generally 2 years or until end of growth when complications arise after such injuries. Comments and discussion on this issue (FUP-time) are requested.

Concerning discussion of results and conclusion:

Following aspects should be taken in account and need to be commented on

- As a surgery-related complication is primarily injury to the growth zone with subsequent growth disturbance due to perforating implants must be mentioned. “Implant failure" is the term used to describe the breakage of screws or wires, is in fact not the fault of the implants and not be attributed to the implants. Implant fracture is usually the result of an unhealed fracture due to insufficient inadequate osteosynthesis, i.e. a mostly surgical problem. The same applies to secondary dislocation occurring early after dislocation occurring early after osteosynthesis or the implant malposition. (Tscherne, Weinberg 2006)

The rate of AVN is mostly dependent on fracture type and amount of dislocation (Mayr et al. 1998 and Pape et al. 1999).

In order it is hard to argue that the findings in this paper are all implant related.

Also, the figures suggest that in this study the findings may also be related to the surgical performance.

The reduction in figure 3 seems inferior to the reduction presented in Figure 2. Further the position of the cranial screw seems to be marginal.

Further in figure 4, the screws reach beyond the growth zone in a 12 year old and is definitely not recommended and a risk factor of AVN and growth disturbances.

The presented figures seem to represent a diversely operated cohort. Therefore, it raises the additional question of who was performing the surgeries? a specialist in paediatric trauma, senior doctor, resident?

Author Response

No abbreviations in the abstract

-Thanks for pointing this out. The abstract section has been corrected as suggested. Abbreviation are no longer in Abstract.

Insert exact p values not only <0.005

-Thank you. We agree. Exact P values are now available throughout the manuscript.

2.6. Error?

-Thank you. The reference has been corrected

Who performed the radiological assessment? Inter -intraobserver?

-Thank you for your question. We take your point. Two experienced pediatric orthopedic surgeons (Y.L. and R.L.) measured all RX parameters independently, and the mean values were used for the statistical analysis; no inter/intra observer assessment was performed for the present study (it is mostly a clinical study not intended to evaluate intra/inter reliability of RX measurements). Thank you for your understanding

Any remaining complications or remaining negative effects (ROM, etc.)?

poor or only good results and remaining difficulties should be described more precisely after this relatively short FUP since recommended Follow-up is generally 2 years or until end of growth when complications arise after such injuries. Comments and discussion on this issue (FUP-time) are requested.

-Thank you very much for your remark. The range of motion (ROM) of the hip was included in functional evaluation. According to Ratliff's criteria, ROM was divided into three grades: good, fair and poor. However, no specific ROM values are provided by Ratliff’s classification system.

This paper mainly discusses the early functional outcome of FNS vs. CCS, and we recognize the short follow-up period (less than 2 years). Relatively short FUP-time is discussed in limitations of the study.

Concerning discussion of results and conclusion:

Following aspects should be taken in account and need to be commented on

- As a surgery-related complication is primarily injury to the growth zone with subsequent growth disturbance due to perforating implants must be mentioned. “Implant failure" is the term used to describe the breakage of screws or wires, is in fact not the fault of the implants and not be attributed to the implants. Implant fracture is usually the result of an unhealed fracture due to insufficient inadequate osteosynthesis, i.e. a mostly surgical problem. The same applies to secondary dislocation occurring early after dislocation occurring early after osteosynthesis or the implant malposition. (Tscherne, Weinberg 2006)

-Thank you. We have reported that Another potential complication is the injury to the growth plate with the implant. However, growth disturbance was not clinically relevant – when occurred – as most patients were approaching skeletal maturity at the time of index surgical procedure. In this paper, “unplanned hardware removal” specifically refers to the lateral irritation of the thigh caused by the backward displacement of the CCS nail tail, requiring a second operation to remove the CCS. However, FNS does not cause lateral irritation of the thigh, and a second operation is generally not needed. The wording “unplanned hardware removal” has been modified with “nail retreat” (as shown in Figure 3-d).

The rate of AVN is mostly dependent on fracture type and amount of dislocation (Mayr et al. 1998 and Pape et al. 1999).

In order it is hard to argue that the findings in this paper are all implant related.

-Thank you. We take your point. The risk factors for AVN suggested by the reviewers have been mentioned in paragraph 5 of discussion. “Previous multivariate analysis of predictive factors for AVN included age, fracture type, amount of dislocation and quality of reduction, but rarely mentioned about fixation methods4,13,22” Cited the latest references (Wang et al. 2019 BJJ; Song et al. 2010 JBJS; Spence et al. 2016 JPO).

Also, the figures suggest that in this study the findings may also be related to the surgical performance.

The reduction in figure 3 seems inferior to the reduction presented in Figure 2. Further the position of the cranial screw seems to be marginal.

Further in figure 4, the screws reach beyond the growth zone in a 12 year old and is definitely not recommended and a risk factor of AVN and growth disturbances.

The presented figures seem to represent a diversely operated cohort. Therefore, it raises the additional question of who was performing the surgeries? a specialist in paediatric trauma, senior doctor, resident?

- Thank you. We take your point. Chapter 2.2. Surgical procedure has emphasized that “All surgical procedures were performed by the same experienced paediatric orthopaedic surgeon (S.C.)”

In paragraph 5 of discussion we have mentioned that “The size of the proximal part of the FNS is smaller than 3 CCSs (Fig. 4), and it may avoid direct damage to the hemoperfusion of the femoral head” and “FNS is easier to insert. Once center guide wire is determined, the rest of the implantation can be completed by the guide tool, resulting in less damage. However, multiple CCSs may require repeated attempts and position adjustment, which is more likely to damage blood supply and lead to AVN”.

Round 2

Reviewer 2 Report

changes are sufficient

This manuscript is a resubmission of an earlier submission. The following is a list of the peer review reports and author responses from that submission.